# Attitudes towards Immigration among Students in the First Year of a Nursing Degree at Universities in Coimbra, Toledo and Melilla

**DOI:** 10.3390/ijerph17217977

**Published:** 2020-10-30

**Authors:** M Idoia Ugarte Gurrutxaga, María Angustias Sánchez-Ojeda, Antonio Segura-Fragoso, María Lucilia Cardoso, Brígida Molina Gallego

**Affiliations:** 1Department of Nursing, Physical and Occupational Therapy, Campus Toledo, University of Castilla-La Mancha, 45004 Toledo, Spain; Maria.Ugarte@uclm.es; 2Department of Nursing, Faculty of Health Sciences, University of Granada, 52005 Melilla, Spain; 3Health Science Institute, Department of Health and Social Affairs, Castilla-La Mancha Government, 45600 Talavera de la Reina, Spain; asegurafr@gmail.com; 4Health Sciences Research Unit: Nursing, School of Nursing of Coimbra, 3000-232 Coimbra, Portugal; luciliacardoso77@hotmail.com; 5FENNSI Group (Functional Exploration & Neuromodulation of the Central Nervous System Group), National Hospital for Paraplegics, 45071 Toledo, Spain; brimoli2010@gmail.com

**Keywords:** attitudes, migration, nursing, students

## Abstract

Increased migration has led to increased prejudice towards immigrant populations. This study aims to analyse attitudes towards immigration among student nurses in three universities, two in Spain and one in Portugal. *Methodology*: A descriptive, transversal, prospective study was carried out among student nurses (*n* = 624), using the Attitude towards Immigration in Nursing scale. *Results*: Nursing students showed some positive attitudes towards immigration, such as that immigrants should have the right to maintain their customs or that immigrants should have free access to healthcare and education, in contrast to some negative attitudes, such as that crime rates have increased due to immigration or that immigrants receive more social welfare assistance than natives. Significant differences in attitudes were revealed between students from the three universities. *Discussion*: Training in transcultural nursing is necessary for all nursing students in order to reduce negative attitudes towards the immigrant population and increase the awareness and sensitivity of future healthcare staff in caring for patients of all backgrounds.

## 1. Introduction

Human migration has increased exponentially. In its World Migration Report 2018, the International Organization for Migration estimated that in 2015 there were almost 244.000.000 international immigrants in the world, i.e., 3.3% of the global population, increasing at a greater rate than anticipated [1].

In 2019, the number of immigrants registered in Spain was 5,036,878, i.e., 9.8% of the total population [2], mainly Moroccans and Romanians. In Melilla, a Spanish city located in the North of Africa, the immigrant population plays an important part in the city’s economic and social life [3]. According to figures published by the Spanish National Institute of Statistics (INE) in 2019 [4], there were 13,266 immigrants living in Melilla, 90% of whom were of Moroccan nationality, and many of them lived in the city in an irregular situation (undocumented). One of the main characteristics of this city was the coexistence of different cultures, although Europeans and Berbers were the largest groups. The Spanish city of Toledo had a population of 84,873 [5], of whom 9533 were born elsewhere, primarily in Morocco and Romania.

In Portugal, according to data from the 2020 Report on Immigration, Borders and Asylum (Portuguese acronym RIFA) prepared by the Portuguese Border and Immigration Service [6], this country was home to 480,300 immigrants holding a residence permit. In the Portuguese city of Coimbra in 2018, 133,940 residents were registered in the city’s municipal census [7]. Of this figure, 6359 were immigrants, primarily Brazilians.

One of the challenges facing today’s societies is managing healthcare in a way that meets the needs of a culturally diverse population. If healthcare professionals are not equipped with the personal and structural tools that allow them to provide healthcare to individuals, families and communities from other cultural backgrounds, this can have a negative impact on the quality of care [8]. Quoting Vilá: “The challenge of providing care in a multicultural city is often not met due to failures in communication, or ignorance, negative attitudes and even rejection of cultural difference” [9].

Numerous studies have shown that the Spanish population holds prejudiced attitudes and negative beliefs as regards the immigrant population [10,11,12], among which is the opinion that there are too many immigrants, or that they are taking jobs away from Spaniards, or that they are responsible for lower healthcare quality, in addition to other anti-immigrant feelings [13,14,15].

As regards the situation in Portugal, according to information in the Migration Observatory Report on Immigrant Integration Indicators [16], the Eurobarometer Survey in 2015 revealed that 64% of individuals surveyed in Portugal considered that discrimination based on ethnic origin was a common phenomenon (the same figure as the average found across the European Union); however, this percentage had increased to 67% in the May 2019 survey (although the average EU figure had dropped to 59%).

This information is relevant insofar as the image of a specific group held by a society is reflected both among student nurses [17,18] and qualified nurses, as revealed by the authors of several studies undertaken in hospital settings [9,19], and similarly in a study on the attitudes of student nurses in Portugal [20]. Such attitudes can have a negative impact on the care process for patients from different cultures [14,15,21,22].

For the reasons stated above, the aim of this study is to shed light on attitudes towards immigration held by student nurses in three universities, two in Spain and one in Portugal.

### Theoretical Framework

Healthcare in which its professionals do not have the cultural skills required, and consequently experience communication difficulties due to language and cultural barriers, can materialize as unequal treatment impacting on the health of the people requiring care [23]. By contrast, culturally competent healthcare leads to improved health outcomes and patient satisfaction [24].

Acquiring awareness of our own cultural values is the first step for training in cultural competence, a term first coined in the late 1980s to address the effects of cultural and linguistic barriers in the interpersonal encounters between healthcare practitioners and clients in health service access and delivery. Cultural competence has been defined as a “set of congruent behaviours, attitudes and policies that come together in a healthcare system, agency or among professionals that enable that system, agency or professions to work effectively in cross-cultural situations” [25].

At the time when the present study was carried out, little training in cultural competence was provided in the health disciplines in the two countries in our study, since students were only trained in transcultural nursing during one term. Bearing in mind the evidence supporting the efficacy of programmes that include cultural competence when dealing with immigrant patients [26], this skill should be included in the nursing degree curriculum gradually and throughout the course of study. Continuous training is even recommended in the clinical sphere to reinforce the development of cultural competence [27]. According to Purnell’s model of cultural competence [28], nurses must be able to manage their own prejudices in order to prevent them from having a damaging effect on the therapeutic process of the patients in their care.

At the beginning of their training, nursing students were found to have a biased view of cultural diversity resulting from socialization in a social context where stereotypical views of and prejudices towards other cultures abounded [29].

The aim of this study was to compare attitudes towards immigration among students in the first year of a nursing degree in three universities: one in Portugal, the College of Nursing in Coimbra (University of Coimbra), and two in Spain, the Faculty of Physiotherapy and Nursing in Toledo (University of Castilla-La Mancha) and the Faculty of Health Sciences in Melilla (University of Granada).

## 2. Materials and Methods

### 2.1. Study Design

A descriptive, transversal study was undertaken to evaluate attitudes towards immigration among future male and female nurses.

### 2.2. Participants

A total of 624 students in the first year of a nursing degree at universities in Coimbra (Portugal), Melilla (Spain) and Toledo (Spain) participated in the study. Participants were selected by means of convenience sampling, the idea being to conduct subsequent research to compare the results of this study on first-year nursing students with the results obtained with third-year nursing students in the same universities. Sample size was calculated to estimate the percentage of students expressing entire agreement with the items in cases where there was a lower degree of agreement (2%). Accepting an alpha risk of 0.95 for an accuracy of ±1.1% in a bilateral comparison for an estimated proportion of 2%, a sample of 624 students was required.

Table 1 shows the characteristics of the study sample, which was organized first by the sex of students (81.7% female and 18.3% male), and then by age, where the average age of students was 19.31 years old and standard deviation 3.34 (min. = 17, max. = 46). Distribution by age showed that 22.6% of students fell into the ≥20 years old bracket and 77.4% were younger than 20 years old; most of them (54.3%) were 18 years old. Concerning the university in which participants were enrolled, it is important to highlight that students in Coimbra account for 43.8% (268 students) of the sample, students in Toledo for 28.4% (175 students) and students in Melilla for 27.9% (174). The study therefore has a higher proportion of students in Coimbra than in Toledo or Melilla.

### 2.3. Study Variables

The study variables collected were sociodemographic variables (gender and age) and the variable “attitude towards immigration”, assessed with the AIN. Although the questionnaire collates all of the sample’s sociodemographic characteristics, the study variables were the university in which the student was enrolled and “attitude towards immigration”, measured using the Attitude towards Immigration in Nursing (AIN) scale.

### 2.4. Instruments

To assess the attitudes among students in the first year of a nursing degree, the Attitude towards Immigration in Nursing (AIN) scale, developed and validated by Antonín and Tomás-Sábado [30], was used in its original Spanish version and in its Portuguese version (translated and culturally adapted) [20].

The AIN scale is a self-administered questionnaire containing 39 items with a Likert response format offering four options, ranging from “Totally Agree” to “Totally Disagree”, scored from four to one or one to four depending on the direction of the item, therefore allowing possible total scores to range from 39 to 156, with higher scores indicating a more favorable attitude towards immigration. That is to say, this summative procedure permits the attitude towards immigration to be quantified on a continuum ranging from the end with the highest score, indicating the most favorable attitude, to the end with the lowest score, indicating a negative attitude with beliefs, emotions and behaviour typical of rejection and prejudice.

### 2.5. Procedure

The study was carried out during the academic courses 2017–2018 and 2018–2019, before the students had starting training in transcultural nursing. The questionnaires were handed out in class to all students in the first semester of the year by the teacher during this semester. They had previously been informed that participation was voluntary and that confidentiality was guaranteed. Each student was allocated a code.

The software IBM SPSS (Statistical Package for the Social Sciences, IBM, Armonk, NY, USA) version 24 (was used for subsequent data analysis. After data had been entered, variables were recoded in the negative direction so that for all of them, higher values indicated a more positive attitude towards immigrants for both sexes of students and lower values a more negative attitude to the same. Analysis of variance (ANOVA) was used, since this method can be used to compare different groups within a quantitative variable. We were thus able to check whether any significant differences existed in the attitudes of students towards immigrants, measured using the AIN scale, according to the university at which they were studying.

### 2.6. Ethical Considerations

The study was undertaken after obtaining authorization from Coimbra College of Nursing. A favorable report was obtained from the Health Sciences Research Unit Ethics Committee: Nursing (UICISA: E) at Coimbra College of Nursing (decision no. 531/10-2018). Furthermore, informed consent was obtained from study participants. Confidentiality and data anonymity were guaranteed when filling in the questionnaire (each student was allocated a code).

## 3. Results

An ANOVA analysis was performed, as explained above, and only statistically significant data were shown according to the university attended by study participants. The data shown in Table 2 below were obtained: Coimbra (268 students), Toledo (175 students) and Melilla (174 students), with 22 of the 39 items in the questionnaire being statistically significant (8 for positive attitudes and 14 for negative attitudes).

For positive attitudes relating to immigrants being allowed to maintain their customs and have a better quality of life (items 3, 5 and 39), students from Coimbra held more positive attitudes than their counterparts in Melilla and Toledo, respectively. Students from Toledo were the most supportive of immigrant children being entitled to free education (item 28), followed by students in Melilla and Coimbra. Students from Melilla were more supportive of immigrants being entitled to free healthcare and education and their own places of worship, and the idea that a higher birth rate was good for the country (items 6, 18, 20), followed by students in Toledo and Coimbra.

Regarding items indicating negative attitudes, students from Coimbra agreed more strongly than students from Melilla and Toledo with the ideas that immigration has led to increased crime rates, that immigrants receive more social welfare assistance and that immigrants should observe the customs of their host country (items 1, 2, 9, 17). A higher percentage of students from Toledo than students from Melilla and Coimbra agreed with the ideas that immigrants were responsible for increased unemployment and healthcare system collapse, that too many resources were earmarked for immigration and that immigrants hindered the country’s development (items 16, 24, 25 and 26). Lastly, for the items that refer to Moroccan immigrants, students from Melilla had more negative attitudes towards them (items 36 and 37) than their counterparts in Toledo and Coimbra.

## 4. Discussion

On the whole, the results obtained in this study show that students have a fairly positive attitude towards the immigrant population. These results match those obtained in other studies, where positive attitudes outweigh negative ones in different populations of healthcare students or practitioners [20,31,32,33,34]. However, we found statistically significant differences in both positive and negative attitudes when observing the university enrolment variable.

Analysis of the items on the scale revealed ideas among students that match those of the general population, such as the anti-immigrant feeling, developed by Betz [13], whereby the population shows negative attitudes towards immigration. This can be observed in the responses to the items “immigrants sometimes receive more social welfare assistance than the autochthonous population itself” and “immigrants take our jobs, leaving many of us unemployed”, where for the first item students in Toledo had a better attitude than students in Melilla, with Coimbra coming in last, and for the second item, students in Toledo had a worse attitude, followed by students in Coimbra with Melilla coming in last.

The notion that immigrants take jobs at the expense of the local population matches the results obtained by Rinken and Velasco [35], during the years when developed countries were experiencing a major economic crisis, and by Sánchez-Ojeda [36], when even after emerging from this crisis, the economic situation had still not recovered.

Other items that may reflect this anti-immigrant feeling are those relating to loss of identity or increased crime rates, for example “immigrants who commit a crime should be deported” and “the arrival of immigrants has led to increased crime”, with the least favorable attitude being held by students in Coimbra, followed by students in Melilla and then students in Toledo; other studies revealed similar data [10,36,37,38,39].

Regarding the healthcare entitlement of the immigrant population, significant differences were observed in responses to the items “the collapse of the health system is largely due to increased immigration” and “in the medium term, the mass arrival of immigrants will cause major healthcare and social problems”, where for both items students in Toledo had a worse attitude, followed by Melilla and Coimbra, in that order. These data match those obtained from a sample of nurses in a district hospital in Melilla [19] and from other hospitals in Spain, where the immigrant population is considered to be a major consumer of healthcare resources [39,40,41,42,43]. The fact that students in Melilla have a worse attitude as regards the immigrant population’s healthcare rights may be explained by the fact that there is only one hospital in this city, which delivers healthcare not only to the city’s residents but also to thousands of Moroccans who cross the border every day specifically to be treated in this hospital [44]. This is a common complaint made by healthcare staff tired of having to deliver care to this population free of charge; this is a very serious issue. Nursing should provide the same attendance to all patients regardless of their origin, so starting training in transcultural nursing and healthcare would be necessary to reduce negative attitudes.

The same explanation could account for the responses to the items “in general, I can’t stand Moroccans” and “I feel unsafe if I see a group of Moroccans”, with students in Melilla having the most negative attitude towards this group of immigrants. Numerous research works and national and European surveys have shown that Moroccans are seen in the poorest light by the European population, similar to populations from other Muslim countries [19,40,45,46]. They were found to be the least socially successful in their host country, because their customs and habits collided with Western ones and they were considered very different to Europeans [47,48,49,50]. Moroccans are the largest immigrant population in Spain and particularly in Melilla, where they make up 90% of the foreign population [5], in contrast to Portugal and specifically Coimbra, and similarly Toledo, which has a very small Moroccan immigrant population. This leads us to posit that perhaps this may be one of the reasons why students in Coimbra and Toledo had a better attitude towards this immigrant profile [6]. This issue should be studied as different cultures live in Melilla, with one of the most numerous being Moroccans, whose traditions and culture are quite rooted in the city. So, the fact that students from that city have developed a negative attitude against that culture highlights the importance of intercultural training at the university in order to reduce those prejudices, as numerous studies have shown that training in transculturality decreases negative attitudes among students and helps to build positive attitudes towards diversity [14,51,52,53,54].

As far as statistically significant positive attitudes, for the following items students in Coimbra showed the best attitude, followed by students in Toledo and then by students in Melilla: “we should guarantee that immigrants are allowed to maintain their customs and cultural practices” and “it is unfair to link the phenomenon of immigration with the increase in crime”. These results match those obtained in a European-wide survey, which ranked Portugal second among European countries as regards having a more positive perception and opinion of immigration [55] because as a country, it does not receive many immigrants, and people who do not come into contact with immigrants are more willing to welcome them, as observed by García-Navarro [45]. This observation could account for the worse attitude among students in Melilla, where there is a larger immigrant population, particularly Moroccans, who cross the border every day to work, beg or go to the city hospital’s A&E department [44].

Ahead of students in Coimbra and Toledo, students at university in Melilla showed a more positive attitude in their opinions regarding the immigrant population having free access to healthcare and education and being entitled to have their own places of worship, and they also viewed the increased birth rate in host countries due to immigration more positively. Other studies revealed similar results [19], in contrast with research where immigration was viewed as something negative for society [35,42,56]; these are quite positive data, which are opposite to the attitude of students in Melilla seeing immigration as the reason for the collapse of the healthcare system.

Although these outcomes are very positive, we must not forget that everybody has the same rights and obligations regardless of their national identity, everybody’s dignity should be respected, and the ethical principles of benevolence not malevolence, justice and independence should be guaranteed [57]. As far as nursing care is concerned, we must remember to deliver it taking cultural characteristics into account [58].

### Limitations

One of the limitations of this study was the selection of first-year students. For future research, we should select students from all four years of the nursing degree in order to analyse if the study of transcultural nursing modifies their attitudes towards immigration. Moreover, we should complete this research with a qualitative study to understand the attitudes of the students.

## 5. Conclusions

Most of the students who participated in this study displayed a fairly positive attitude towards the immigrant population, although some of the areas analysed revealed a fairly negative attitude towards immigrants, with first-year nursing degree students showing prejudice and stereotypical views. It should be noted that at the time of conducting this study, these undergraduates had not yet studied the Transcultural Nursing module. This could have an influence on their responses.

Students at the faculty in Melilla showed more negative attitudes than those in Toledo and Coimbra in their responses to the items “In the medium term, the mass arrival of immigrants will cause major healthcare and social problems”, “I can’t stand Moroccans” and “I feel unsafe when I see a group of Moroccans”. This outcome may be related to the sociodemographic situation in each of the cities where the universities in our study are located; the immigrant population in Melilla is far larger.

It is necessary to continue training in transcultural nursing for all nursing students as well as to increase cultural competence and to identify and work on negative attitudes towards immigration until their disappearance, in such a way as to increase the sensitivity and awareness of future nursing professionals who will care for patients of different nationalities and situations.

## Figures and Tables

**Table 1 ijerph-17-07977-t001:** Sociodemographic characteristics of the study sample.

		No.	%
**Sex**	Male	114	18.3%
	Female	510	81.7%
**Age**	17 years old	41	6.6%
	18 years old	339	54.3%
	19 years old	103	16.5%
	≥20 years old	141	22.6%
**University**	Coimbra	273	43.8%
	Toledo	177	28.4%
	Melilla	174	27.9%

**Table 2 ijerph-17-07977-t002:** Degree of agreement with items in the Attitude towards Immigration in Nursing (AIN) scale organized by university where students are enrolled.

Items in the Attitude towards Immigration in Nursing (AIN) Scale	Coimbra (268)			Toledo (175)			Melilla (174)			*p*
	% Totally Agree	CI 95%	Mean	% Totally Agree	CI 95%	Mean	% Totally Agree	CI 95%	Mean	ANOVA
**1. Foreigners who commit a crime should be deported**	15.7	(11.53–20.51)	2.51 (0.94)	9.1	(5.31–14.34)	2.82 (0.93)	14.4	(9.51–20.36)	2.56 (0.98)	**0.002**
**2. The arrival of immigrants has led to increased crime rates**	2.6	(1.03–5.19)	3.13 (0.78)	0.6	(0,01–3.12)	3.10 (0,79)	2.3	(0.63–5.75)	2.89 (0.82)	**0.006**
**3. We should guarantee immigrants the right to maintain their customs and cultural practices**	54.9	(48.83–60.74)	3.42 (0.72)	36.4	(29.25–43.70)	3.22 (0.69)	47.7	(40.08–55.10)	3.41 (0.62)	**0.004**
**5. We should increase our efforts to improve immigrant quality of life**	45.4	(39.41–51.35)	3.32 (0.73)	32.4	(25.53–39.62)	3.07 (0.77)	36.8	(29.61–44.17)	3.18 (0.73)	**0.003**
**6. Whether legal or illegal, immigrants should have free and unrestricted access to healthcare and education**	30.4	(25.00–36.10)	2.86 (0.97)	37.9	(30.68–45.19)	3.10 (0.84)	77.6	(26.91–41.23)	2.98 (0.90)	**0.056**
**9. Immigrants sometimes receive more social welfare assistance than the autochthonous population itself**	37.1	(31.37–43.02)	1.84 (0.80)	29.5	(22.91–36.67)	2.20 (1.01)	23.1	(17.06–29.96)	2.26 (0.94)	**0.000**
**10. Immigrants are generally less concerned about their personal hygiene**	0.7	(0.08–2.63)	3.49(0.65)	1.7	(0.35–4.84)	3.37 (0.76)	2.9	(0.94–6.54)	3.13 (0.84)	**0.000**
**16. Immigrants take our jobs, leaving many of us unemployed**	6.3	(3.68–9.78)	2.76(0.84)	13.6	(8.88–19.39)	2.76 (1.04)	5.2	(2.39–9.54)	3.05 (0.90)	**0.002**
**17. Immigrants should observe the customs of our country**	11.7	(8.15–16.08)	2.54(0.89)	6.9	(3.59–11.60)	2.86 (0.89)	8.6	(4.90–13.74)	2.86 (0.90)	**0.000**
**18. We should allow immigrants to have places of worship where they can practise their beliefs and religions**	50.2	(44.09–56.07)	3.34(0.77)	42.6	(35.20–50.01)	3.20 (0.83)	51.7	(44.03–59.03)	3.43 (0.65)	**0.02**
**20. Immigration and the resultant increase in the birth rate is positive for countries struggling to maintain their demographic balance**	30.04	(25.00–36.10)	2.94(0.90)	31.6	(24.78–38.86)	3.01 (0.86)	43.4	(35.85–50.81)	3.22 (0.81)	**0.004**
**23. The collapse of the health system is largely due to increased immigration**	2.2	(0.82–4.77)	3.27(0.75)	8	(4.41–12.91)	2.91 (0.93)	6.9	(3.63–11.73)	2.71 (0.86)	**0.000**
**24. Too many resources are earmarked for immigrant care**	1.1	(0.23–3.19)	3.33(0.71)	6.9	(3.61–11.67)	2.94 (0.86)	2.9	(0.944–6.57)	2.96 (0.79)	**0.000**
**25. A European citizen living in Spain/Portugal generates more benefits for our society than an immigrant from Africa, Asia or South America**	1.1	(0.23–3.19)	3.61(0.67)	5.8	(2.82–10.37)	3.02 (0.94)	4	1.63–8.06)	3.10 (0.86)	**0.000**
**26. Immigrants and gypsies hinder the country’s development**	0.7	(0.09–2.62)	3.31(0.70)	1.1	(0.13–3.99)	3.44 (0.72)	0.6	(0.01–3.14)	3.60 (0.60)	**0.000**
**28. The children of immigrants, whether legal or illegal, are entitled to free education**	39	(33.13–44.88)	3.06(0.94)	56.3	(48.58–63.37)	3.38 (0.82)	49.1	(41.46–56.52)	3.27 (0.85)	**0.001**
**33. In truth, we make use of immigrants to do the most precarious jobs**	9.2	(6.03–13.22)	2.29(0.98)	18.2	(12.78–24.55)	2.64 (0.94)	31.6	(24.78–38.86)	2.93 (0.95)	**0.000**
**34. In the medium term, the mass arrival of immigrants will cause major healthcare and social problems**	4.4	(2.29–7.52)	2.93(0.82)	11.4	(7.08–16.91)	2.70 (0.94)	9.2	(5.34–14.42)	2.67 (0.92)	**0.004**
**35. If I am in a position to choose, I prefer not to sit next to an immigrant on public transport**	0.4	(0.01–2.03)	3.92(0.34)	0	0	3.84 (0.40)	0	0	3.84 (0.44)	**0.031**
**36. In general, I can’t stand Moroccans**	0.4	(0.009–2.02)	3.97(0.24)	0.6	(0.014–3.14)	3.78 (0.52)	0.6	(0.01–3.14)	3.80 (0.53)	**0.000**
**37. I feel unsafe when I see a group of Moroccans**	1.1	(0.23–3.17)	3.69(0.60)	5.2	(2.41–9.59)	3.24 (0.89)	8	(4.47–13.05)	3.11 (0.96)	**0.000**
**39. It is unfair to link the phenomenon of immigration with the increase in crime**	63	(56.97–68.51)	3.48(0.78)	50	(42.33–57.35)	3.24 (0.89)	56.3	(48.61–63.48)	3.30 (0.92)	**0.006**

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
