# Peer review of "Attitudes towards Immigration among Students in the First Year of a Nursing Degree at Universities in Coimbra, Toledo and Melilla"

_ijerph, 2020, doi:10.3390/ijerph17217977_

Round 1
Reviewer 1 Report
I appreciate the opportunity to review this article and I congratulate the authors for the proposed objective, as well as for the methodological development of the study.
Here are some suggestions for improvement to be considered:
In the abstract, the authors must describe the exact results obtained in their cross-sectional study, as well as show the most relevant conclusions reached with the proposed research.
Most of the bibliographic references are older than 5 years, it would be interesting in the theoretical framework and in the discussion to expand with the bibliography of the last 5 years on the importance of the cultural competencies of professionals and students. I am sure it would provide a more rigorous view of the study context.
I hope that proposals can contribute to improve the article.
Author Response
REVIEWER 1
I appreciate the opportunity to review this article and I congratulate the authors for the proposed objective, as well as for the methodological development of the study.
Here are some suggestions for improvement to be considered:
In the abstract, the authors must describe the exact results obtained in their cross-sectional study, as well as show the most relevant conclusions reached with the proposed research.
Modified results: lines 21 to 24.
Modified conclusions: lines 25 to 27
Most of the bibliographic references are older than 5 years, it would be interesting in the theoretical framework and in the discussion to expand with the bibliography of the last 5 years on the importance of the cultural competencies of professionals and students. I am sure it would provide a more rigorous view of the study context..
Both the Introduction and the theoretical framework have updated the bibliographic references
I hope that proposals can contribute to improve the article.
Thank you very much for your contributions, we hope you like them.

Reviewer 2 Report
The paper is a plain description of attitudes to immigrants in groups of students planning to go into health care.
As such the information is valuable and can be used for teaching and education of students who will get very important positions in the future primary health care.
The language is plain and needs some minor revisions, probably in-house. The methods are adequate as they use a simple questionnaire for assessment.
A very interesting aspect is the young age. The gender difference is as expected, with mainly female participants.
The difference between locations is not so important, but the change in attitude when the immigration is increasing is interesting.
All in all, this report deserves publication after the needed language washing.
Author Response
REVISOR 2
The paper is a plain description of attitudes to immigrants in groups of students planning to go into health care..
As such the information is valuable and can be used for teaching and education of students who will get very important positions in the future primary health care.
The language is plain and needs some minor revisions, probably in-house. The methods are adequate as they use a simple questionnaire for assessment.
A very interesting aspect is the young age. The gender difference is as expected, with mainly female participants.
The difference between locations is not so important, but the change in attitude when the immigration is increasing is interesting.
All in all, this report deserves publication after the needed language washing.
We sincerely appreciate your feedback.
We hope that the proper language

Reviewer 3 Report
Dear Authors,
Thanks for allowing me to read your valuable work. In my opinion, this manuscript could be strengthened in multiple ways and I will explain section by section, what I would suggest to improve.
Overall, the manuscript has to be edited by an editor to improve grammar and sentence structure.
Abstract:
- The first sentence of your abstract makes a global statement but does not reflect the situation in Spain or Portugal, nor does it give an explanation why this would be relevant in first-year nursing students.
- Why were there two schools from Spain and one from Portugal. What makes them comparable? Mellila is on the African continent in Morocco, Toledo is in close proximity to Madrid, so far away from any coast, and Coimbra is also far away from any boarder with Morocco.
- I would suggest naming your methodology as cross-sectional descriptive study design.
- When was the study conducted?
- Be more specific when you say fairly positive, what are they positive about and what are the regional differences. The same is true for the negative perceptions.
- Who do you need to work with and especially how would you do that to change their negative perceptions.
- Please order keywords alphabetically
Background:
- Please start your description of immigration with the situation in Spain and Portugal and explain where most migrants come from. Also explain, why these migrants come to either country. The situation in Mellila is unique as it is within the borders of Morocco, so Morrocons can just come for a day to receive healthcare, they are not migrants.
- Please provide a table with the total population of each city, the number of migrants, which is different from foreign born people in 2018 or 2019, and the number of hospitals in each city.
- Explain how health care is delivered in both countries to citizens and non-citizens.
- What are the negative attitudes and prejudices against foreigners? You never say that specifically.
- Then there is a big jump, no explanation why this is important for nursing students, no discussion about the gaps in the literature and what this study would add.
- The theoretical section lacks much clarity and a graph to explain how the concepts relate to each other. Please describe who coined "cultural competence and what its major concepts are. Without that it is very difficult to understand the use of the scale.
- You say aquiring cultural awareness of our own cultural values is the first step for training in cultural competence, but if I understand you right they have a pretty good grasp on their own negative values. So explain what you mean by that. How does that lead to cultural sensitivity, the ability to work with vulnerable populations, and lower their own negative attitudes towards foreigners?
- Would you explain why there is little training in cultural competency in both countries?
- Who is Purnell and what is her model? Did it influence your thinking?
- How do you come to the conclusion that students often have in both countries stereotypic views of foreigners?
- Does the study focus on attitudes towards immigration or immigrants and why is this important? Your study aim is not clear about that. It tells the what and the where, but does not answer the why?
Methods:
- Please clarify in your aim if you mean immigration in general or their attitudes towards immigrants or migrants.
- Your study is cross-sectional descriptive - nothing more.
- How did you determine the sample size, how many UG nursing students does each program have. How were the students approached to participate?
- Please explain your sample size calculations better. What do you mean by lower degree of agreement (2%). With what? Please provide reference.
- Did this account for incomplete responses? How many did you have?
- What year was this research done?
- The demographics belong into the result section.
- The single sentence paragraph for study variables needs to be reworded, it is unclear what the authors want to say here.
- No reliability and validity reported on the study scale. Also the year when there scale was developed and translated is missing. What are the key concepts of the scale? Does the scale have sub scales and sub scale scores?
- Has there be a factor structure established with the items of the scale?
- How high was the refusal rate?
- How was the data handled after the students completed the questionnaire?
- Who entered the data into a database?
- Why was there not a multiple regression analysis done to determine the predictors of positive and negative attitudes towards immigration?
- A simple ANOVA is not enough to make meaningful inferences.
- You only show ethics approval from one site, what happened in the other sites?Why is not all data shown and only the significant one.It would help with documenting the performance of the scale.
- The overall results of the scale should be presented first followed by an ANOVA, then followed by three multiple regressions for each site.
Discussion:
- The discussion is not structured around what was found, what was not found, and what needs to be done in the future. Some statements of whether students do or not hold attitudes about immigrants, contradict each other and need to be revised. This discussion also does not provide an in-depth discussion of the causes or future opportunities of the attitudes towards immigrants.
- The additional analysis would also allow to connect these findings to other parts of Southern Europe that see a high influx of immigrants.
- The notion of whether immigrants in fact take the jobs of people from Spain or Portugal was not supported by any real data, and fact checking and reporting would be needed.
- The same is true for the collapse of the healthcare system. What numbers are available to link this to the number of immigrants in those cities. All of which are quite low.
- There are no limitations reported for this study.
Conclusion"
- What is meant by the term "display a fairly positive attitude" it is very subjective and means what?
- Issues such as the unique location and situation of Mellila are discussed in the conclusion and they belong into the discussion. They have never been brought up before and come as a surprise.
- In the conclusion two issues are raised that could be easily done with this data. Gender comparisons and age comparisons without any justification why this was not done.
- A transcultural nursing course is mentioned but it is unclear if that course exists in all three sites and when this course is being taught.
References:
The references need some editing. Not consistantly formatted.
Author Response
REVISOR 3
Thanks for allowing me to read your valuable work. In my opinion, this manuscript could be strengthened in multiple ways and I will explain section by section, what I would suggest to improve.
Overall, the manuscript has to be edited by an editor to improve grammar and sentence structure.
We hope that, after reviewing the language, the grammar and sentence structure are suitable for the article
We greatly appreciate your contributions and we hope that all modifications are necessary.
Abstract:
- The first sentence of your abstract makes a global statement but does not reflect the situation in Spain or Portugal, nor does it give an explanation why this would be relevant in first-year nursing students.
Words limitation in the abstract affects its content. The named aspect is approached with accurate precision in the Introduction section.
- ¿ Why were there two schools from Spain and one from Portugal. What makes them comparable? Melilla is on the African continent in Morocco, Toledo is in close proximity to Madrid, so far away from any coast, and Coimbra is also far away from any boarder with Morocco.
This research has been made since the authors worked in one of the three faculties where the data were obtained. We met on a research stay and we believed this questionnaire could be suitable to be done on each nursing school to find out if Nursing students had the same attitudes or not.
- I would suggest naming your methodology as cross-sectional descriptive study design.
We agree
- When was the study conducted?
In the academic year 2017-2018 and 2018-2019
- Be more specific when you say fairly positive, what are they positive about and what are the regional differences. The same is true for the negative perceptions.
It has been modified
- Who do you need to work with and especially how would you do that to change their negative perceptions?
The phrase has been improved in the Abstract (Discussion part)
- Please order keywords alphabetically
keywords have been ordered
Background:
- Please start your description of immigration with the situation in Spain and Portugal and explain where most migrants come from. Also explain, why these migrants come to either country. The situation in Melilla is unique as it is within the borders of Morocco, so Moroccans can just come for a day to receive healthcare, they are not migrants.
The information has already been organized.
Moroccans must pay for health care when they attended, but the reality isn’t like that. There isn’t any agreement on free healthcare for Moroccans.
- Please provide a table with the total population of each city, the number of migrants, which is different from foreign born people in 2018 or 2019, and the number of hospitals in each city.
In the text it has been specified that we refer to immigrant population. Hospitals from each city where the study was carried out don’t provide any relevant information for the object of our study.
- Explain how health care is delivered in both countries to citizens and non-citizens.
In Spain, all people with their regular administrative situation are guaranteed health care within the framework of the National Health System model. In Portugal, three months of residence in the country is required in order to receive health care.
- What are the negative attitudes and prejudices against foreigners? You never say that specifically.
In lines 57-60, it is specified that the population has negative attitudes and prejudices towards the immigrant population.
- Then there is a big jump, no explanation why this is important for nursing students, no discussion about the gaps in the literature and what this study would add
The aim of this type of research is to evaluate a hypothetic situation to act in the future, in our part as teachers of Transcultural Nursing, to be able to act with our students to reduce prejudices
- The theoretical section lacks much clarity and a graph to explain how the concepts relate to each other. Please describe who coined "cultural competence and what its major concepts are. Without that it is very difficult to understand the use of the scale.
The scale that we have used in the study focuses on attitudes towards immigration, it is not a scale to measure cultural competence. The term cultural competence appears in the literature around the 80s, coming from anthropology. The concept is generated in the United States and England as a result of the large number of people from different parts of the world that made evident the cultural shock in the approach to health-disease generated between immigrants and health professionals. Since then, there is a large production of scientific literature referring to the cultural competence developed by health professionals who have graduated from the human sciences, from which explanations have emerged to understand the interactions of the elements of culture and their influence on health.
- You say aquiring cultural awareness of our own cultural values is the first step for training in cultural competence, but if I understand you right, they have a pretty good grasp on their own negative values. So, explain what you mean by that. How does that lead to cultural sensitivity, the ability to work with vulnerable populations, and lower their own negative attitudes towards foreigners?
In line 86-87 it is better explained.
- Would you explain why there is little training in cultural competency in both countries?
Lines 86-87. Transcultural Nursing is only studied for one semester in the first year of the degree, when it should be a cross-cutting theme throughout the 4 years of study.
- Who is Purnell and what is her model? Did it influence your thinking?
Purnell's model of cultural competence (1995) can be used by all professions related to health care in a multidisciplinary mode. This model is applicable in all professional practice situations and in any context. According to Purnell, cultural competence is the adaptation of care to the culture of the patient. We consider that this model perfectly frames our study. Purnell’s model has a lot influence in our study.
- How do you come to the conclusion that students often have in both countries stereotypic views of foreigners?
Some studies carried out previously, highlight that students develop negative attitudes towards immigration similar to the population in general.
Lines 60-69
- Does the study focus on attitudes towards immigration or immigrants and why is this important? Your study aim is not clear about that. It tells the what and the where, but does not answer the why?
Nursing should involve positive attitudes towards any group, as it can influence care. It seemed very important to us to make a comparison of the three Faculties, because the authors have already studied attitudes towards immigration on their own.
Méthods:
- Please clarify in your aim if you mean immigration in general or their attitudes towards immigrants or migrants.:
The aim of our work is to know the attitude towards immigration of the students of the first year of Nursing, attitude towards the immigrant population.
- Your study is cross-sectional descriptive - nothing more.
Our study was aimed at studying in a precise moment some attitudes (measured by validated scale). We interviewed a sample of some voluntary students.
- How did you determine the sample size, how many UG nursing students does each program have. How were the students approached to participate?.
It was carried out a convenience sampling. This sampling was non-probabilistic, casual o incidental and in order to reach the suitable sample size, all students who, were previously informed, and voluntarily decided to participate in the study were included and confidentiality was guaranteed.
- Please explain your sample size calculations better. What do you mean by lower degree of agreement (2%). With what? Please provide reference
EPIDAT version 3.1 software package was used to estimate the sample size. The different assumptions have been based on our own data from a previous pilot test. The 2% is the percentage of students who chose "Total agreement" among the 4 available options that are: "Total Agreement", "Moderate Agreement", "Moderate Disagreement" and "Total Disagreement”.
- Did this account for incomplete responses? How many did you have?
The incomplete answers were eliminated, this was an exclusion criterion for our study. The surveys in which all questions were answered correctly were selected
- What year was this research done?
The research was carried out in the years 2017, 2018 and 2019, in the different faculties. Being in Portugal in 2017-2018 and in Spain in 2018-2019
- The demographics belong into the result section.
The number of participants according to sex, age groups and different faculties
- The single sentence paragraph for study variables needs to be reworded, it is unclear what the authors want to say here.
Corrected
- No reliability and validity reported on the study scale. Also, the year when there scale was developed and translated is missing. What are the key concepts of the scale? Does the scale have sub scales and sub scale scores?
In order to evaluate the attitude of students, the EAIE, developed and validated by Antonín-Martín and Tomás-Sábado (2004), was used in the Spanish (original) and Portuguese (translated and culturally adapted in 2017) versions. The EAIE is a self-ad-ministered Likert-type questionnaire consisting of 39 items with Likert type 4-option response, ranging from total agreement to total disagreement, with a score system from four to one or from one to four, depending on the direction of the item, so that the possible total scores range from 39 to 156, in which higher scores indicate more favorable attitudes towards
- Has there be a factor structure established with the items of the scale?
The objective of this work is not to validate the "Attitude towards Immigration in Nursing (AIN)" scale, but to describe the attitudes by using that scale. Therefore, no factor analysis is appropriate.
The scale items scored from four to one or from one to four according to the direction of the different items. Higher scores, more favorable attitudes towards immigration. It is a summative procedure to quantify the attitude towards immigration.
- How high was the refusal rate?
All students answered the questionnaire voluntarily.
- How was the data handled after the students completed the questionnaire?
The normality and degree of symmetry of the neuropsychological variables were checked by using graphical methods and tests of normality Kolmogorov-Smirnov Test.
The qualitative variables are presented by percentages and the quantitative variables by centralization and dispersion measures (mean and standard deviation) and 95% confidence intervals. The association between qualitative variables was analyzed using the Chi square test. The statistical analysis was performed by using IBM SPSS for Windows, version 24.0.
We used ANOVA, because it helps us to compare several groups in a quantitative variable, checking if the attitudes of the students towards the immigrants, measured through the total sum of the items of the Attitudes Scale, differ between the students according to the faculty of origin (Melilla, Coimbra, Toledo).
- Who entered the data into a database?
First, the questionnaires were distributed collectively among the students (first semester of the course), each teacher who handled the questionnaires entered the data in the database. Later, the person in charge of the methodological part, merged these databases. Then, we rercoded the variables with negative sense and finally the data was analyzed with the IBM SPSS for Windows, version 24.0.
- Why was there not a multiple regression analysis done to determine the predictors of positive and negative attitudes towards immigration?
Because it was not an aim of this study. This study simply tried to describe the attitudes of students towards the immigrant population.
- ¿ A simple ANOVA is not enough to make meaningful inferences.
We don’t understand that comment, but if the intention is to compare the attitudes in the three faculties, the ANOVA is a suitable test to make this comparison.
- You only show ethics approval from one site, what happened in the other sites?Why is not all data shown and only the significant one.It would help with documenting the performance of the scale
The study was first set in Coimbra and then the approval of the Ethical Committee was obtained, in the others centers (Melilla and Toledo) the approval was requested to the Committees in charge these matters. In each center it was managed in a particular way according to the guidelines of the Faculty.
We carried out different analysis according to the student’s Faculty in order to study our objective, but finally in the article only the most significant results have been reflected. It was a matter of space since the table occupied a significant number of the sheets allowed. Anyway we have the whole analysis in case you ask for it.
- The overall results of the scale should be presented first followed by an ANOVA, then followed by three multiple regressions for each site.
It could be done but it is a laborious job. We would have to do a multiple regression for each item, that is 39*3=117 regressions. This is very difficult to represent graphically, besides the limitation of words which determines its content.
Discusión:
- The discussion is not structured around what was found, what was not found, and what needs to be done in the future. Some statements of whether students do or not hold attitudes about immigrants, contradict each other and need to be revised. This discussion also does not provide an in-depth discussion of the causes or future opportunities of the attitudes towards immigrants.
In the discussion, we have outlined the possible reasons of the results found.
- The additional analysis would also allow to connect these findings to other parts of Southern Europe that see a high influx of immigrants.
We appreciate your suggestion that opens a new line of research for us.
- The notion of whether immigrants in fact take the jobs of people from Spain or Portugal was not supported by any real data, and fact checking and reporting would be needed.
These data are prejudices that general population has developed about migrations, which are obviously not real.
- The same is true for the collapse of the healthcare system. What numbers are available to link this to the number of immigrants in those cities. All of which are quite low.
Definitely, it would be very appropriate an analysis that will relate the number of immigrants to the saturation level of the health system. We will take your reflection into account for future research.
- There are no limitations reported for this study.
Study limitations realized in lines 267-272.
“Conclusión"
- What is meant by the term "display a fairly positive attitude" it is very subjective and means what?
With that term “display a fairly positive attitude” we understand that most students have a feeling in favor of the immigrant population.
- Issues such as the unique location and situation of Melilla are discussed in the conclusion and they belong into the discussion. They have never been brought up before and come as a surprise.
It has been modified and included in the discussion, as you suggested
- In the conclusion two issues are raised that could be easily done with this data. Gender comparisons and age comparisons without any justification why this was not done.
This was due to the extension of the article
- A transcultural nursing course is mentioned but it is unclear if that course exists in all three sites and when this course is being taught.
The contents related to intercultural care are taught in the three sites where the research has been carried out, in the first year of nursing training.
References:
The references need some editing. Not consistently formatted.
The references are corrected.

Round 2
Reviewer 3 Report
Hello,
Thanks for allowing me to review these revisions. The revision has clarified much of the initial concerns. I still see English edits that should be made, for example:
Instead of "as regards" it would be better to say "Concerning, regarding, about....
The entire document should be written in past tense as this study happened in the past.
Table 2 has formatting issues and the sentence in the legend:
"Own elaboration of the authors" is unclear. what does this mean?
All nationalities should be capitalized.
It is not bereberes but rather Berbers.
It still don't understand the relevance of the transcultural training. Was the study done before or after the students have received their first training? Would the findings then reflect the changed attitudes based on the training, or pertaining to attitudes more similar to the general population.
In the discussion, this issue is not raised again, of what are the plans for addressing transcultural training between the two countries and three different schools.
Because the situation of Melilla is so unique as a Spanish enclave in Africa sharing a border with Morocco, this should also be discussed more in detail, of what are the plans to assure that nurses and nursing students deliver culturally competent care. The people seeking care are more day visitors than migrants. They don't stay in Melilla permanently if I understand this correctly. I am still struggling to see how this is t he same as immigrants that come into another country permanently.
Three terms are used, migrants, immigrants, inmigrants, do they all mean the same, none have been defined in the background.
Did the other two universities not require any ethics approval? I see only one university ethics committee say yes to the study?
The discussion falls short to recommend future studies to explore this topic more and too explain better the role of nurses.
Author Response
Comments and Suggestions for Authors
Hello,
Thanks for allowing me to review these revisions. The revision has clarified much of the initial concerns. I still see English edits that should be made, for example:
Instead of "as regards" it would be better to say "Concerning, regarding, about..
Reply: Ok. It has been changed in the text; we hope it´s ok
Instead of "as regards" it would be better to say "Concerning, regarding, about..
Reply: Ok. It has been changed in the text; we hope it´s ok
Table 2 has formatting issues and the sentence in the legend
Reply: Done
"Own elaboration of the authors" is unclear. what does this mean?
Reply: Done
All nationalities should be capitalized..
Reply: Done
It is not bereberes but rather Berbers.
Reply: Done
It still don't understand the relevance of the transcultural training. Was the study done before or after the students have received their first training? Would the findings then reflect the changed attitudes based on the training, or pertaining to attitudes more similar to the general population.
Reply: This has been described in the procedure part, the questionnaires were passed before starting training in transcultural nursing: “the questionnaires were handed out in class to all students in the first semester of the year by the teacher during this semester….”, in order to know what were the attitudes of the student.
In the discussion, this issue is not raised again, of what are the plans for addressing transcultural training between the two countries and three different schools.
Reply: In the nursing degree of three faculties, training in transcultural nursing area is included in level one.
Because the situation of Melilla is so unique as a Spanish enclave in Africa sharing a border with Morocco, this should also be discussed more in detail, of what are the plans to assure that nurses and nursing students deliver culturally competent care. The people seeking care are more day visitors than migrants. They don't stay in Melilla permanently if I understand this correctly. I am still struggling to see how this is t he same as immigrants that come into another country permanently..
Reply: Moroccans not only come to be treated in the hospital, but also thousands of them stay in the country in an irregular situation (undocumented). this has been described in lines 39 and 40. also in the discussion lines 221-223.
Three terms are used, migrants, immigrants, inmigrants, do they all mean the same, none have been defined in the background.
Reply: The term is already unified, only we used immigrant
Did the other two universities not require any ethics approval? I see only one university ethics committee say yes to the study?.
Reply: The require to pass the questionnaire in melilla has been added in the paper. in toledo, as we had a favorable resolution from coimbra (ethics committee), we only needed dean`s authorization
The discussion falls short to recommend future studies to explore this topic more and too explain better the role of nurses..
Reply: It have been added five references.
lines 239-241.